

# Prevalence and factors associated with *Schistosoma mansoni* infection among primary school children in Kersa District, Eastern Ethiopia

Hussen Aliyi[1], Mohammed Ahmed[2], Tesfaye Gobena[3], Bezatu Mengistie Alemu[4], Hassen Abdi Adem[3] and Ahmedin Aliyi Usso[5]

[1] East Hararghe Health Bureau, Harar, Ethiopia
[2] School of Medical Laboratory Sciences, College of Health and Medical Science, Haramaya University, Harar, Ethiopia
[3] School of Public Health, College of Health and Medical Science, Haramaya University, Harar, Ethiopia
[4] School of Public Health, Saint Paul Hospital Millennium Medical College, Adis Ababa, Ethiopia
[5] School of Nursing and Midwifery, College of Health and Medical Science, Haramaya University, Harar, Ethiopia

Corresponding author
Mohammed Ahmed,
mameahmed129@gmail.com

## ABSTRACT

**Background**. Schistosomiasis is a neglected tropical disease and an important parasite negatively impacting socio-economic factors. Ethiopia's Federal Ministry of Health targeted the elimination of schistosomiasis infection in school-aged children by 2020. However, *Schistosoma mansoni* still affects approximately 12.3 million school-aged children in Ethiopia. Although the study was conducted in some regions of the country, previous studies were conducted on urban school children and were limited to the burden of infection. Overall, there is a lack of information about schistosomiasis in eastern Ethiopia, particularly among school children. Therefore, this study aimed to assess the prevalence and factors associated with *Schistosoma mansoni* infection among primary school children in Kersa district, Eastern Ethiopia.

**Methods**. A cross-sectional study was conducted among 419 school children using systematic random sampling from April 10 to May 09, 2019. The stool samples were collected and examined using the Keto-Katz method. A structured and pretested questionnaire was used to collect data from participants. Data were entered using Epi-Data version 3.1 and analysed using SPSS version 24. A bivariable and multivariable logistic regression analyses were used to identify factors associated with *Schistosoma mansoni* infection. P-value < 0.05 and adjusted odds ratio (AOR) (95% confidence interval (CI)) were used to identify statistically significant associations.

**Results**. This study's overall prevalence of *S. mansoni* was 19.4% (95% CI [16–23]). Absence of the latrines in household (AOR = 2.35, 95% CI [1.25–4.38]), swimming in the river (AOR = 2.82, 95% CI [1.33–5.88]), unprotected water sources (AOR = 3.5, 95% CI [1.72–7.10]), irregular shoe wearing habits (AOR = 2.81, 95% CI [1.51–5.23]), and water contact during cross of river (AOR = 2.192; 95% CI [1.113–4.318]) were factors independently associated with *S. mansoni* infection.

**Conclusion**. *Schistosoma mansoni* infection remains a public health problem in the study area. Using a latrine in each household, using protected water, wearing shoes regularly, and reducing water contact were necessary to control *Schistosoma mansoni* infection.

# INTRODUCTION

Schistosomiasis is the neglected tropical infectious disease that affected more than 220 million people worldwide in 2017, of which approximately 90% of them live in Sub-Saharan Africa (*WHO, 2020*). According to WHO definitions, schistosomiasis is an acute and chronic disease caused by trematode of the genus *Schistosoma* (*WHO, 2002*). *S. mansoni, S. haematobium*, and *S. japonicum* are the primary causes of human schistosomiasis, and less commonly *S. mekongi* and *S. intercalatum* (*Caldas et al., 2008*; *CDC, 2018*).

Schistosomiasis is the second most common tropical disease after malaria and infects over 219.9 million people globally, of whom approximately 54.7% are school-age children (*WHO, 2018*). Most commonly, schistosomiasis infects people living in poor communities that lack access to clean water and adequate sanitation (*Cohee et al., 2020*; *Karagiannis-Voules et al., 2015*). Intestinal schistosomiasis caused by *S. mansoni* can be manifested as acute pain accompanied by bloody diarrhea and intestinal ulcers, while chronic infections lead to liver enlargement, liver fibrosis, portal hypertension, and hematemesis (*King & Dangerfield-Cha, 2008*; *van der Werf, Bosompem & De Vlas, 2003*). Schistosomiasis has been linked to long-term chronic inflammatory reactions and can lead to anemia and malnutrition, which in turn results in stunted growth, poor academic performance, low work productivity, and persistent poverty (*King, 2010*). *S. mansoni* infection can also increase the risk of malnutrition in children, leading to cognitive impairment and reduced physical function (*Colley, 2015*).

Schistosomiasis has been brought under control in many countries but remains a serious public health problem with severe economic consequences in areas where hygiene and control measures are inadequate and the majority of the population lives in poverty (*Molyneux, Savioli & Engels, 2017*; *Sady et al., 2013*). According to WHO reports at least 75% of school-age children were protected from infection at the end of 2010 (*WHO, 2010*), yet it did not succeed.

In Ethiopia, the Federal Ministry of Health (FMoH) has set a goal of eliminating schistosomiasis infection in school children through mass drug administration of praziquantel (PZQ) as a main strategy for control of schistosomiasis by 2020 (*Mengitsu et al., 2016*). However, schistosomiasis remains a major health problem in developing countries (*Engels et al., 2002*), affecting approximately 12.3 million school-age children in Ethiopia (*MoHEthipia, 2016*). Furthermore, the previous studies were conducted only on urban school children (*Assefa, Dejenie & Tomass, 2013*) and were limited to the burden of infection (*Alebie et al., 2014*). The schistosome parasites develop and multiply into the infective cercariae within the snail hosts and are released into the water bodies by the snails (*Abe et al., 2018*). people become infected when the parasite's larval stages, released by freshwater snails, penetrate the skin after coming into contact with infested freshwater, causing long-term morbidity and, in severe cases, death (*Anto et al., 2013*;

*Deribe et al., 2012*). Urogenital schistosomiasis caused by *Schistosoma haematobium*, is known to be endemic in numerous lowland parts of Ethiopia and poses significant public health problems to schoolchildren. Ethiopia, after mapping the disease's distribution (2013–2015), initiated a school-based mass deworming program in schoolchildren to eliminate schistosomiasis across the country since 2015 (*Deribew et al., 2022*).

This study was crucial as the study site was suitable because there was a river between the schools and their residential area. Also, there was an irrigational activity for the agricultural products around the river, which is suitable for *S. mansoni* intermediate host. An effort has been made to map the epidemiology of *S. mansoni* infections in different geographical regions of the country. It is not possible to conclude that the disease's geographic distribution has been completely mapped out, as recent findings suggest a rise of new transmission foci that may be related to the expansion of irrigation initiatives and human movement (*Alemayehu & Tomass, 2015*).

Overall, there is a lack of information about schistosomiasis in eastern Ethiopia. Therefore, this study aimed to assess the prevalence and factors associated with *Schistosoma mansoni* infection among primary school children in Kersa district, eastern Ethiopia.

The findings will strengthen existing knowledge and support policymakers in coming up with effective *S. mansoni* infection prevention strategies by identifying the factors that contribute to the persistence of the problems.

## METHODS AND MATERIALS

### Study design, area, and period

A cross-sectional study was conducted in Kersa District, Oromia, Eastern Ethiopia located 493 KM away from Addis Ababa the capital city of Ethiopia, and 33 KM from Harar city. The Kersa district has three climate zones with altitudes ranging from 1,600 to 3,200 m above sea level. Rain generally falls in two seasons; the main rainy season is from June to September and the short rainy season is from March to May. The annual rainfall varies between 1,400 to 1,900 mm. The district has a warm climate, with yearly temperatures ranging between 18 °C and 26 °C. The district has 31 primary schools and 27,861 students enrolled in 2018/19 (*Kersa District Health Office, 2017*).

### Populations

The source population includes all children enrolled in grades 1 to 8 of the 31st elementary school in Kersa district during the 2018/2019 academic year. The study population was all children attending the selected primary schools. Those who have recently transferred from a school outside the Kersa district within the past six months, who were critically ill, and who were treated for *S. mansoni* within the past three months were excluded from the study.

### Sample size determination and sampling

The sample size was calculated by Epi-Info version 7.2 using a single population proportion formula for estimating the prevalence of *S. mansoni* infections and two population proportions formulas for factors associated with *S. mansoni* infection. The following
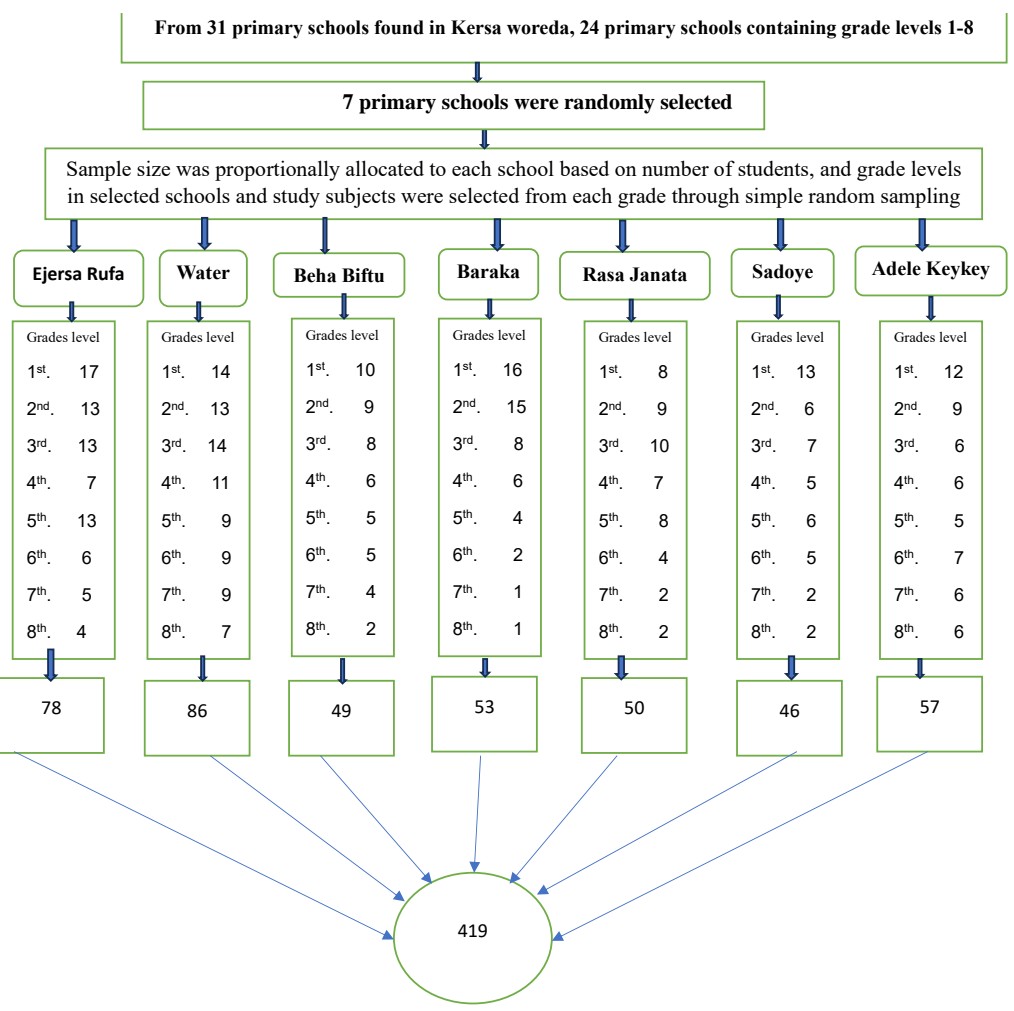

**Figure 1** Diagrammatic representation of the sampling technique of study participants, proportional allocation of schools children by grade level; and selection of study subject through simple random sampling from seven schools included in the study, in Kersa district, 1st, 2nd, to 8th represents the grade levels of the students.

assumptions were used to compute the sample size for the prevalence of *S. mansoni* infection: 45% proportion of *S. mansoni* infection (*Amsalu, Mekonnen & Erko, 2015*), 95% confidence level, 5% margin of error, and 10% non-response rate and accordingly, a minimum of 419 participants required to conduct the study.

Seven primary schools were randomly selected from 31 primary schools found in Kersa district. The sample size was distributed proportionally to each selected primary school based on the number of students enrolled in Kersa Primary School during the 2018/2019 academic year. Study participants were selected through simple random sampling from each grade and class of the selected primary school by using class rosters as a sampling frame after proportionally allocating the number of participants to each selected primary school (Fig. 1).

## Data collection method

A pretested structured questionnaire adapted from different literature (*Alemayehu et al., 2017*; *Assefa, Dejenie & Tomass, 2013*; *Hailu et al., 2018*; *Worku et al., 2014*) was used. The questionnaire includes sociodemographic characteristics, environmental factors, personal factors, and *S. mansoni* infection. The questionnaire was prepared in English, then translated into the local Afan Oromo language, and re-translated back to English. Six trained data collectors were deployed to collect data under the supervision of two public health officers. The principal investigator conducted training for the data collectors and supervisors.

## Stool sample collection, processing, and examination

Written consent was obtained from the children's parents/guardians. To ensure the accuracy of the information, the youngsters participating in the study were interviewed in their native language. For students who were unable to answer the questions correctly, guardians were contacted through the school principal.

Students selected for the study were given instructions on how to collect stool samples, as well as how to use an applicator stick, toilet paper, and a clean, labeled plastic container. Immediately upon receipt of stool samples, labeling, quantity, timing, and collection method were reviewed for each sample. Two slides were prepared from a single stool sample and examined two times. Intestinal schistosomiasis eggs were examined in stool samples using a method known as the Kato-Katz technique (*WHO, 2023*).

## Kato-Katz method

The Kato-Katz technique (a modification of the direct smear procedure) is especially useful for field surveys for the identification of *S. mansoni* infection (*WHO, 1994*). Because of its low cost, short sample preparation time, simple handling, the need for only basic equipment, and simplicity, it is the main method of schistosomiasis diagnosis and nearly the only one currently in use in research laboratories, however, it's not a sensitive diagnostic technique in areas of low prevalence (*Barbosa et al., 2017*; *Bärenbold et al., 2017*; *Martin & Beaver, 1968*).

In the present study, two Kato-Katz thick smears were prepared from different parts of the single stool sample using a template of 41.7 mg (*WHO, 1994*). In the Kato-Katz technique, the eggs of *S. mansoni* were concentrated by straining the feces through the mesh screen to remove the large pieces of debris. Then the filtered sample was transferred into a template hole and placed on a microscope slide until the hole was filled. Then after, the template was carefully removed, and the remained sample was covered by a cellophane strip which was presoaked in a solution of glycerine and malachite green that made easy detection of eggs (*RABELLO, 1992*). Kato-Katz was used to prepare stool smears on slides for microscopic examination at the Water and Kersa Health Center.

## Quality control

To assure the accuracy and reliability of the results, a questionnaire was pretested, and the principal investigator conducted a two-day training session for the data collectors and supervisors on how to conduct the interview and collect stool samples. The questionnaire
was checked daily for accuracy and completeness, and the principal investigator closely monitored the entire data collection process. To ensure reliable laboratory results, qualified laboratory technologists conducted laboratory tests using the Kato-Katz technique standard operating procedures. Principal investigator cross-checked specimens to improve the accuracy of laboratory results. To avoid observer bias, the smears were examined independently by competent laboratory technologists. Additionally, 10% of both positive and negative Kato-Katz smears were randomly selected and re-examined by two independent medical laboratory experts who were unaware of the primary results. In case of discordant results between the two laboratory personnel, the principal investigator reviewed the samples and determined the conclusive results.

## Ethical consideration and consent

Institutional Health Research Ethical Review Committee of Haramaya University, College of Health and Medical Sciences approved the protocol of the study by this reference number (IHRERC/069/2018). Formal permission was obtained from the Zonal Health Office, the District Health Office, and each selected school. The child's parents/guardians provided informed, voluntary, written consent. The children gave their verbal consent. Verbal consent was also obtained from the children. Participants were interviewed at a separate location after being told that the collected information would be kept confidential and would not be accessible except to authorized persons. Children who tested positive for *S. mansoni* during the study period have been linked to Water and Kersa Health Center for treatment.

## Data processing and analysis

After checking for completeness, the collected data were entered using Epi-Data version 3.1 and analyzed using SPSS version 24 (IBM, Armonk, NY, USA). Descriptive statistics such as frequencies, tables, and figures were used to present the data. Bivariate and multivariable logistic regression analyses were applied to identify factors associated with *S. mansoni* infection. Adjusted odd ratio (AOR) with 95% CI was used to determine the association between the dependent and independent variables. The statistical significance was declared at $P < 0.05$.

## RESULTS

### Socio-demographic characteristics

A total of 413 primary school children participated in the study with a response rate of 98.6%. The participant age ranges from 6 to 18 years old with a mean ($\pm$SD) age of 11.33 ($\pm$SD 2.72) years. The majority of participants (240 (58.1%)) were found in the age group of 10–14 years old, followed by 118 (28.6%) in the age group 7–9 years and 55 (13.3%) 15–18 years old.

Regarding the gender of the participants, about 285 (69 %) were males. Among the participants' educational levels, 271 (65.6%) students were in grades 1–4, and 142 (34.4%) students were in grades 5–8. Two hundred eight three (68.4%) of the participants' fathers and about 342 (82.6%) of their mothers had no formal education. Of the family occupations,

**Table 1 Socio-demographic characteristics of primary school children in Kersa district, Oromia, Eastern Ethiopia, from April 10 to 30, 2019 (n = 413).**

| Characteristics | Frequency (number) | Percent (%) |
| --- | --- | --- |
| **Sex** | | |
| Male | 285 | 69.0 |
| Female | 128 | 31.0 |
| **Age** | | |
| 6–9 | 117 | 28.3 |
| 10–14 | 240 | 58.1 |
| 15–19 | 56 | 13.6 |
| **Grade** | | |
| 1–4 | 271 | 65.6 |
| 5–8 | 142 | 34.4 |
| **Religion** | | |
| Muslim | 407 | 98.5 |
| Orthodox | 5 | 1.3 |
| Protestant | 1 | 0.2 |
| **Father's education status** | | |
| Literate | 130 | 31.5 |
| Illiterate | 283 | 68.5 |
| **Mother's education status** | | |
| Literate | 71 | 11.2 |
| Illiterate | 342 | 82.8 |
| **Fathers' Occupation** | | |
| Farmer | 381 | 92.3 |
| Merchant | 26 | 6.3 |
| Employee | 4 | 1.0 |
| Others | 2 | 0.50 |
| **Mothers' Occupation** | | |
| Housewife | 376 | 91.0 |
| Merchant | 35 | 8.5 |
| Employee | 2 | 0.5 |

381 (92%) of the school children's fathers were farmers, 26 (6.3%) merchants, and 4 (1.0%) government employees (Table 1).

## Environmental and personal characteristics

Regarding environmental factors, the majority, 243 (58.8 %) of the participants use water from rivers and streams for drinking and other domestic purposes. Nearly half of the participants (46.97%) did not have access to restrooms and instead relied on open fields. About 77 (18.6%) of children had practiced regular hand washing, 183 (44.2%) had washed their hands inconsistently, and 154 (37.2%) had not washed their hands at all after defecation. The majority of children (259, 62.71%) bathe in rivers and streams, while nearly four-fifths (77.48%) of schoolchildren wash their clothes in rivers or ponds. One hundred sixty-two (39.2%) youngsters had a habit of swimming when crossing the river to

**Table 2** Environmental and personal characteristics of primary school children in Kersa district, East Hararghe, Eastern Ethiopia, 2019 ($n = 413$).

| Characteristics of the student's | Frequency (number) | Percent (%) |
|---|---|---|
| **Source of water for households** | | |
| Pipe | 170 | 41.2 |
| River/Stream | 243 | 58.8 |
| **Latrine in the households** | | |
| Yes | 219 | 53.0 |
| No | 194 | 47.0 |
| **Latrine utilization** | | |
| No | 196 | 47.40 |
| Yes | 217 | 52.60 |
| **Received health education** | | |
| Yes | 18 | 4.3 |
| No | 395 | 95.7 |
| **Swimming in the river** | | |
| Yes | 162 | 39.2 |
| No | 251 | 60.8 |
| **Bathing in the river/stream** | | |
| Yes | 259 | 62.7 |
| No | 154 | 37.3 |
| **Washing clothes in the river** | | |
| Yes | 320 | 77.5 |
| No | 93 | 22.5 |
| **Crossing of the river** | | |
| Yes | 110 | 26.6 |
| No | 303 | 73.4 |
| **Shoes wearing habits** | | |
| Regular | 224 | 54.2 |
| Irregular | 189 | 45.8 |
| **Participation in irrigation farm** | | |
| Yes | 138 | 33.4 |
| No | 275 | 66.6 |

school and from school to their family houses. Over half (54.24%) of the schoolchildren wore shoes. One in every three (33.4%) school children's families use irrigated agricultural fields as a source of income (Table 2).

## Prevalence of *S. mansoni* Infection

The prevalence of *S. mansoni* infection was 19.4% (95% CI [16–23]). The prevalence of *S. mansoni* by the school among the studied schools ranges from 11.53% in Ejersa Rufa Primary School to 36.73% in Beha Biftu Primary School. Prevalence of *S. mansoni* concerning age group revealed that the majority, 50 (62.5%) of *S. mansoni* infections were in the age range of 10 to 14 years, 19 (23.75%) were in 15 to 18 years, and 11 (13.75%) were in 6–9 years old children, respectively (Fig. 2).

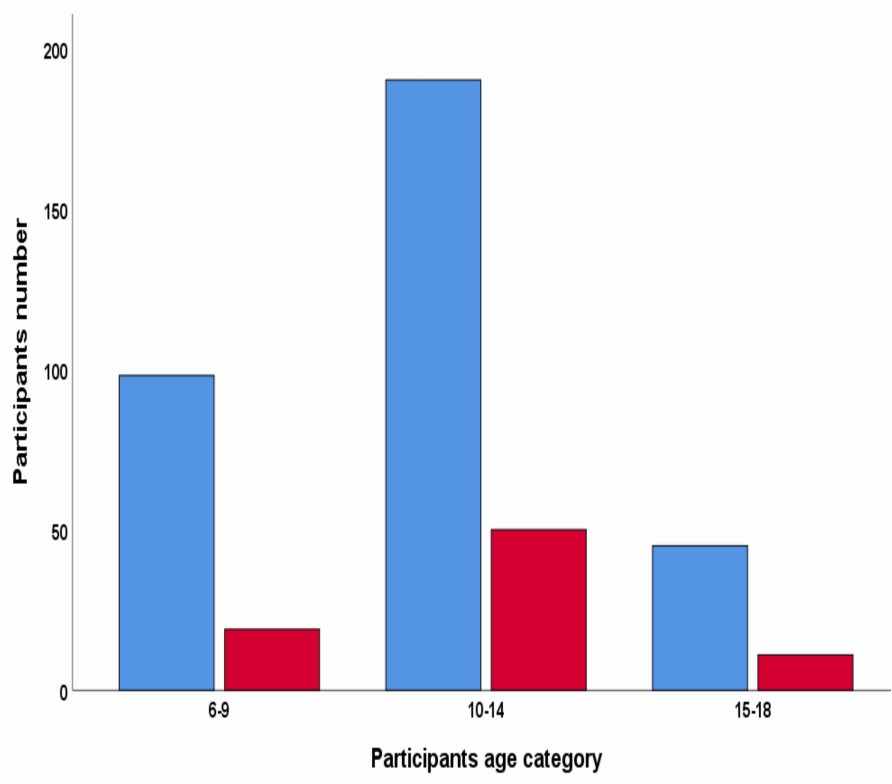

**Figure 2** *Schistosoma mansoni* **infection by Kato-Katz results of stool samples.** The red color indicates positive Kato-Katz results of the stool sample. The blue color indicates negative by Kato-Katz results of the stool.

## Factors associated with *S. mansoni* infection

In binary logistic regression latrine, water source, swimming in rivers or ponds, bathing in rivers or ponds, washing clothes in rivers, family irrigation activities, the habits of wearing shoes, and crossing rivers or streams on the way to school were all found to be significantly associated with *S. mansoni* infection. However, in multivariable logistic regression lack of a latrine at home, swimming in the river, bathing in the river, the habit of wearing shoes, crossing the river to get to school, and sources of water remained significantly associated with *S. mansoni* infection.

The children whose families did not have a latrine at home were 2.34 times more likely infected with *S. mansoni* than those whose families had a latrine at home (AOR = 2.34; 95% CI [1.25–4.38]). In contrast, *S. mansoni* infection was 2.8 times higher in children who swam in the river (AOR = 2.80; 95% CI [1.33–5.88]) compared to those who didn't swim totally. The odds of children not wearing the shoes were 2.81 times more likely infected with *S. mansoni* than children with the habits (AOR = 2.81; 95% CI [1.51–5.23]). The children who crossed the river to get to the school had higher odds of *S. mansoni* infection than their counterparts (AOR = 2.19, 95% CI [1.11–4.32]). Ultimately, the proportion of children whose families used rivers as a source of drinking water were more likely to

**Table 3 Factors associated with *S. mansoni* infection among primary school children in Kersa district, Eastern Ethiopia, 2019 (*n* = 413).**

| Characteristics 1 | *S. mansoni* test result | | COR (95% CI) | AOR (95% CI) |
|---|---|---|---|---|
| | Positive, n (%) | Negative, n (%) | | |
| Household latrine available | | | | |
| Yes | 26(11.9) | 193(88.1) | 1 | 1 |
| No | 54(27.8) | 140(72.2) | 2.86(1.71, 4.79) | **2.34 (1.25, 4.38)**[*] |
| Swimming in the rivers/ponds | | | | |
| No | 19 (7.8) | 229 (92.3) | 1 | 1 |
| Yes | 61 (37.0) | 104 (63.0) | 7.07(4.02,12.4) | **2.80 (1.33, 5.88)**[*] |
| Bathing in the river/ponds | | | | |
| No | 2(1.3) | 151(98.7) | 1 | 1 |
| Yes | 78(30.0) | 182(70.0) | 32.36(7.82, 133.9) | **19.21(3.10,119.22)**[**] |
| Washing in the river/ponds | | | | |
| No | 6(6.1) | 93(93.9) | 1 | 1 |
| Yes | 74(23.6) | 240(76.4) | 4.78 (2.01, 11.4) | 0.50 (0.13, 1.89) |
| Shoes wearing habit | | | | |
| Always | 26(11.6) | 198(88.4) | 1 | 1 |
| Sometimes | 54(28.6) | 135(71.4) | 3.05 (1.817, 5.1) | **2.81 (1.51, 5.23)**[**] |
| Participate in irrigation farm activities | | | | |
| No | 32(11.6) | 243(88.4) | 1 | 1 |
| Yes | 48(34.8) | 90(65.2) | 4.05 (2.44, 6.73) | 1.13(0.56, 2.27) |
| Crossing the River Road to school | | | | |
| No | 36(11.9) | 267(88.1) | 1 | 1 |
| Yes | 44(40.0) | 66 (60.0) | 4.94 (2.95, 8.29) | **2.192 (1.113, 4.32)**[**] |
| Household source of water | | | | |
| Pipe | 17(10.0) | 153(90.0) | 1 | 1 |
| River | 63(25.9) | 180(74.1) | 3.15 (1.77, 5.61) | **3.495 (1.72, 7.10)**[**] |
| Total | 80 | 333 | | |

Notes.
[*] $P < 0.05$.
[**] $P < 0.01$.
AOR, Adjusted OR; CI, Confidence Interval. Bold value indicating for significance.

become infected with *S. mansoni* than those who used tap water (AOR = 3.49; 95% CI [1.72–7.10]) (Table 3).

## DISCUSSION

Schistosomiasis infections are among the most prevalent neglected tropical diseases that affect people, especially schoolchildren who live in underdeveloped countries. Due to the impact on children, the World Health Organisation (WHO), World Bank, and other United Nations agencies are working to control schistosoma through regular programs of targeted school-based mass drug administrations and deworming of both schistosomiasis and soil-transmitted helminths (*WHO, 2002*; *WHO, 2003*; *Mengitsu et al., 2016*).

According to the current study, the prevalence of *S. mansoni* infection among school children was 19.4% with 95% CI [16–23]. Lack of latrine at home, swimming in the river, bathing in the river, wearing shoes, crossing the river to get to school, and the source of drinking water were statistically associated with *S. mansoni* infection.

This study finding was consistent with the study done in Kersa Eastern Ethiopia (16.2%) (*Gemechu et al., 2023*), Northwest Ethiopia, 20.6% (*Essa et al., 2013*), Kenya, (16.3%) (*Handzel et al., 2003*), 19.8% from Ghana (*Anto et al., 2013*), and White Nile, Sudan, (19.8%) (*Tamomh et al., 2018*).

However, this finding was higher than the study conducted across the country including Gamo Gofa and South Omo, southern Ethiopia (14%) (*Girma et al., 2018*), northwest Ethiopia (15.2%) (*Wubet & Damtie, 2020*), Ejaji, southwest Ethiopia (12.9%) (*Ibrahim et al., 2018*), Bahir Dar, northwest Ethiopia (8.0 %), (*Hailu et al., 2018*), and similarly from different part of Latin America and African countries, Nigeria (8.9%) (*Awaki et al., 2016*), 1.5% from Bamako, Mali (*Dabo et al., 2015*), and Brazil, 14.4% (*Barbosa et al., 2006*).

The reason for the high prevalence of *S. mansoni* infection in the current study compared to other findings might be due to differences in proximity to water bodies, long-time endemicity in the area, and differences in water contact behaviour of school children to infected water for different purposes. In this study 77.5% of school children washed clothes in the river/ stream, 39.2% swimming in the river, and 62.7% bathing in the river/stream (Table 2). In addition to this, the difference might be due to the nature of the climate (difference in altitude and temperature) of the study area, since the infection is influenced by climatic changes as the climates affect the transmission of this infection by providing the soil with the necessary moisture and warmth for snail development (*Brooker, Clements & Bundy, 2006*; *Hailu et al., 2018*; *Mas-Coma, Valero & Bargues, 2009*).

On the other hand, this finding was lower than the study conducted in Mekelle, northern Ethiopia 23.9% (*Assefa, Dejenie & Tomass, 2013*), Fincha'a, west Ethiopia (53.2%) (*Mekonnen et al., 2014*), Lake Tana, northwestern Ethiopia (34.9%) (*Hailegebriel et al., 2021*), Wolaita, southern Ethiopia (58.6%) (*Alemayehu et al., 2017*) and Sanja, northwest Ethiopia (89.9%) (*Worku et al., 2014*). This finding was also lower than the study done in Uganda (27.8%) (*John et al., 2008*), northwestern Tanzania, 64.3% (*Mazigo et al., 2010*), and 69% from Nyanza province Kenya (*Samuels et al., 2012*).

The low prevalence of *S. mansoni* infection in this finding might be due to the expanded national deworming program in 2015 among school children that may contribute to decrements in the prevalence of infections (*Mengitsu et al., 2016*; *Mwandawiro et al., 2019*; *WHO, 2015*). In addition, the variation among the studies might be due to the difference in water contact behaviour of the children, variation of sample size & study period; endemicity of the infection, source of water in those areas, and geographic variation of the study area (*Mekonnen et al., 2014*; *Hailegebriel et al., 2021*; *Mazigo et al., 2010*).

Among factors associated with *S. mansoni* infection, children without latrines in their homes were 2.34 times more likely to contract *S. mansoni* than those whose families have latrines at home. This finding was similar to the study conducted in Southern Ethiopia (*Tadege & Shimelis, 2017*), and the finding from Haike primary schools, in North-East Ethiopia (*Feleke et al., 2017*). This could be due to a lack of latrines in the study area, forcing most people to defecate in open fields, especially near water sources, contaminating the water with human waste and allowing *S. mansoni* to spread when schoolchildren come into contact with cercaria-infested water bodies.

The current study found that children who drank river/stream water were 3.5 times more likely to be infected with *S. mansoni* than those who drank tap water. This finding was in line with the studies conducted in Mekelle, Northern Ethiopia (*Assefa, Dejenie & Tomass, 2013*), and Kisantu, Democratic Republic of Congo (*Kumbu, Makola & Bin, 2016*). This could be because rivers and streams are frequently contaminated by human and animal excreta, freshwater becomes contaminated in with snails that carry schistosomes, then the snail release infectious forms of parasite known as cercariae into the water, resulting in parasite transmission when schoolchildren's skin comes in contact with contaminated freshwater infested with active cercaria while fetching the water for drinking or other household purposes.

In addition, schoolchildren who had not worn shoes regularly were 2.8 times more likely infected with *S. mansoni* than those who had worn shoes regularly. This finding was similar to the study done in Gamo Gofa and South Omo (*Girma et al., 2018*), Jiga, Northwest Ethiopia (*Worku et al., 2014*), Mana, South West Ethiopia (*Bajiro & Tesfaye, 2017*), and, Zarima, Northwest Ethiopia (*Alemu et al., 2011*). This might be due to walking barefoot exposing children to cercariae-infected water when they cross the river to school and back home from school. Therefore, the concerned body should educate the community on how to prevent children and themselves from schistosomiasis and soil-transmitted diseases including the importance of wearing shoes.

The current study showed that the odds of *S. mansoni* infection were 2.2 times higher in schoolchildren who had physical contact with water while crossing the river or stream than in children who did not have physical contact (AOR = 2.2; 95% CI [1.11–4.32]). This finding was in line with the study conducted in different parts of the Ethiopian regions including Tigray, Northern Ethiopia (*Assefa, Dejenie & Tomass, 2013*), Jimma, Southwest Ethiopia (*Bajiro & Tesfaye, 2017*), and Jiga, Northwest Ethiopia (*Worku et al., 2014*).

This might be due to the location of the rivers/streams containing *S. mansoni* cercariae which the children came into contact with when crossing the river, as well as the school's proximity to the river, which allows the children to frequently contact the water and encounter them to bathe and swim together for recreation or other purposes, resulting in direct contact with cercariae released by snail, the intermediate host transmitting schistosomiasis (*Brooker, 2007*). As a result, the responsible bodies should take action to prevent the spread of *S. mansoni*, and health education should be offered to promote awareness among school children about the risks of infected water contact, such as swimming in a river, which can lead to *S. mansoni* infections.

This finding reveals that school children who had swum in the river/ponds had 2.8 times higher odds of being infected with *S. mansoni* than those who did not swim in the river/ponds. Similar findings reported from Wolaita, Southern Ethiopia (*Alemayehu et al., 2017*), Northwest Ethiopia (*Hailu et al., 2018*), Zarima Town, Northwest Ethiopia (*Alemu et al., 2011*), and World Health Organisation estimates confirmed that school-aged children swimming in infested water are more likely to become infected with *S. mansoni* (*WHO, 2014*; *WHO, 2012*). It could be because swimming exposes the entire body of the school children to cercariae-infected water bodies, which facilitates transmission (*WHO, 2012*).

The present study indicated that the water contact habits of school children affirmed that swimming and bathing habits in rivers were significantly associated with higher odds of infection with *S. mansoni*. This finding was in agreement with the studies conducted in Northwest Ethiopia (*Hailu et al., 2018*), Jimma Zone, South West Ethiopia (*Bajiro & Tesfaye, 2017*), and Ejaji town, Central Ethiopia (*Ibrahim et al., 2018*). The main reason for this significant association might be due to the proximity of the school to the river bodies occupied with infected snails which directly release infective stage cercaria into the water.

Overall, from this finding, we believe that targeted intervention for at-risk population groups with the already initiated deworming programs and high-burden spots, together with the widespread implementation of integrated snail control measures is crucial in mitigating the burden of Schistosomiasis locally and beyond to accomplish the WHO's global aim of eliminating neglected tropical diseases by 2030.

### Strengths and limitations of the study

The standard Kato-Katz procedures were employed to detect eggs, and experienced laboratory professionals checked the slide to ensure the parasite was not missed. Furthermore, unlike prior studies that focused on urban areas, this study included different schools in rural areas. The study's limitation was that it employed a single stool sample, which may not address day-to-day and intra-stool variations of eggs. In addition, we did not perform egg intensities.

## CONCLUSION AND RECOMMENDATION

This finding suggested that *S. mansoni* infection among schoolchildren remained a public health issue. The primary school children were found to be more at risk of *S. mansoni* infection when there were no latrines at home, irregular shoe wear, unprotected water, and frequent contact with river water. Community-based health education is essential to promote latrine use in homes, wearing shoes, and helping the community obtain a safe drinking water source is crucial. The children's various actions and behaviors are related to their helminthic infection. It is critical to provide health education regularly to prevent risky behaviors and water-based activities that could expose children to this infection and other helminths. So, it is recommended to control schistosomiasis and other Helminthes by developing strategies to control snails, ensuring widespread provision of WASH, and conducting regular deworming campaigns as preventive chemotherapy.

### Abbreviations

| | |
|---|---|
| **AOR** | Adjusted Odds Ratio |
| **EDHS** | Ethiopian Demographic Health Survey |
| **NTD** | Neglected Tropical Disease |
| **WASH** | Water Sanitation and Hygiene |
| **WHO** | World Health Organization |
| **DALY** | Disability-Adjusted Life Year |
| **GBD** | Global Burden of Diseases |

## ACKNOWLEDGEMENTS

We thank all study participants, data collectors, and supervisors. We also want to thank East Hararghe Health Bureau, the Kersa District Health office, and all data collectors and laboratory technicians at the respective schools of the study.

### Funding

The study was funded by Haramaya University as part of an MSc study to Hussen Aliyi. The fund were only to facilitate data collecting. The funders had no role in study design, data collection and analysis, decision to publish, or preparation of the manuscript.

### Grant Disclosures

The following grant information was disclosed by the authors:
Haramaya University.

### Competing Interests

The authors declare there are no competing interests.

### Author Contributions

- Hussen Aliyi conceived and designed the experiments, performed the experiments, analyzed the data, authored or reviewed drafts of the article, and approved the final draft.
- Mohammed Ahmed conceived and designed the experiments, performed the experiments, analyzed the data, prepared figures and/or tables, authored or reviewed drafts of the article, and approved the final draft.
- Tesfaye Gobena conceived and designed the experiments, performed the experiments, analyzed the data, authored or reviewed drafts of the article, and approved the final draft.
- Bezatu Mengistie Alemu conceived and designed the experiments, performed the experiments, analyzed the data, authored or reviewed drafts of the article, and approved the final draft.
- Hassen Abdi Adem conceived and designed the experiments, performed the experiments, analyzed the data, authored or reviewed drafts of the article, and approved the final draft.
- Ahmedin Aliyi Usso conceived and designed the experiments, performed the experiments, analyzed the data, prepared figures and/or tables, authored or reviewed drafts of the article, and approved the final draft.

### Human Ethics

The following information was supplied relating to ethical approvals (i.e., approving body and any reference numbers):

The Institutional Health Research Ethical Review Committee of the College of Health and Medical Sciences, Haramaya University approved this research (IHRERC/069/2018).

## Data Availability

The raw data is available in the Supplemental Files.

## Supplemental Information

Supplemental information for this article can be found online at http://dx.doi.org/10.7717/peerj.17439#supplemental-information.

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
