# Peer review of "Prevalence and factors associated with Schistosoma mansoni infection among primary school children in Kersa District, Eastern Ethiopia"

_PeerJ, doi:10.7717/peerj.17439_

## Round 0.1 · original submission · Major Revisions

The review process has been completed, and three highly qualified referees have provided thorough feedback, which you can find at the bottom of this letter. However, there are some concerns that you need to address in your resubmission. I agree with the reviewers and stress that you must be committed to addressing the points raised with the utmost precision and incorporating all the necessary changes in the manuscript to be submitted. It is important to improve the English language to make it clear and comprehensible to readers.

**Language Note:** The Academic Editor has identified that the English language must be improved. PeerJ can provide language editing services - please contact us at [email protected] for pricing (be sure to provide your manuscript number and title). Alternatively, you should make your own arrangements to improve the language quality and provide details in your response letter. – PeerJ Staff

Reviewer 1 ·

Basic reporting

The paper presents informative data on S. mansoni infections and associated risk factors in Ethiopia. The authors have produced a good paper and the data is important to publish to show that S. mansoni is still a public health problem in Ethiopia. The paper should be improved with the addition of more detail and correction of some mistakes.

See additional comments for evaluation and changes needed

Experimental design

See additional comments for evaluation and changes needed

Validity of the findings

See additional comments for evaluation and changes needed

Additional comments

Here is a list of comments and changes needed
General
Check spelling and grammar throughout. Many times species names are not in italics.

Introduction
54 – leeches is the wrong term to use for trematodes
55 – there are more than one species that cause schistosomiasis
60 – you already talk about global burden in the previous paragraph
76 – be more specific for schistosomiasis – there seems some over lap with STH in your paper – talk specifically about treatment of schistosomiasis with PZQ (chemotherapy). Add information about schistosomiasis control programmes which, is usually conducted via MDA of PZQ.
82 – add more of the life-cycle / transmission. There is a lack of information or association with the snail host in the paper. State what the snail host is in Ethiopia.
You should also mention urogenital schistosomiasis in relation to Ethiopia.

Methods
128 – explain what you mean by S. mansoni infection in the questionnaire e.g. have they been screened, treated or have any knowledge of past infections
141 – add more information on the KK. How many slides, how many times the slide was read, was egg intensity recorded?
144 – KK is not a sensitive method for S. mansoni particularly at low intensity infections. Explain that and also discuss in the limitations.
146 – you talk about intensity of infection but you do not report on this at all. Also, 43 grams of stool is a lot and egg intensities are usually calculated as number of eggs per 1g. I think there is a mistake here as you say 42 grams of stool were collected but how would you measure that.
Also, add information on when the study was done and the data gathered.
148 – what quality control was done on the KK slides.
161 – you need to add more information on how the participants were treated.

Results
177 – about is too vague – you cannot have about 285 males. Also, why not report on females too?
187 – several of your risk factors are strongly correlated to STH. E.g hand washing and wearing shoes. You should explain more clearly what relevance these have to schistosomiasis.
193 – what about intensity data?

Discussion
235 – this is related to STH not schistosomiasis
240 – explain more about the deworming – is that focused on STH or schistosomiasis or both – what treatments are used
253 – explain why drinking water is a risk. Infection does not occur from ingestion.
280 – infection does not occur through direct contact with the snails. Ad more detail of snails here.
297 – I do not see KK analysis as the strength. The combined data is the strength. Limitations include lack of intensity data also
304 – change susceptible to more at risk
311 – explain what the other helminths are
You should also mention urogenital schistosomiasis in relation to Ethiopia.

Figures and tables
A map of the villages in relation to rivers, streams and ponds is needed.
Figure 1 – needed some editing to male it ready for publishing. E.g. why have grade 1-8 in each box as it is the same for all.
Figure 2 – more information is needed in the legend. On the graph what do you mean by frequency? Refer to laboratory results as KK results. You could also show the data more clearly. One bar for the whole age category divided by those infected and not infected. That would make it more meaningful.
Each table seems to be labelled 1 ?
First table – this is a very difficult table to understand. Rethink how to show the data and add more information to the legend. This table could go in supplementary material
Last table – add over all pos and neg and break down by age to the table

·

Basic reporting

The English language in the manuscript should be substantially improved so that an international audience can understand the message being communicated. The author should consider professional English language editorial services to markedly improve the manuscript. Grammatical issues and unclear statements and phrases appear in several places in the manuscript, including lines 53, 73-73, 83, 105-16, 158-159, 160, 186-187, 191, 195, 198, 209-210, 232, 237, 241, 261, 275, 279, 290-291, 306.

The introduction and background show context of the study. This can however be improved further by avoiding misleading terms such as leeches as the cause of schistosomiasis in line 54. The author also omits other causes of human schistosomiasis, only mentioning one causative species in line 55. The global number of cases of schistosomiasis is repeated in lines 51 and 60, this is unnecessary.
The authors can also improve this section by giving information if there have been ongoing disease control interventions among school kids in Kersa district and for how long.

The author has cited extensively relevant literature throughout the manuscript. There are several key references with findings relevant for this study that were not cited, including one school children in Kersa Woreda in eastern Ethiopia by Gemenchu et al., 2023. This study, though focusing on Anemia in school children, did examine parasite infections including S. mansoni and found a prevalence of 16.2%. Several good systematic reviews and meta-analyses have been on S. mansoni prevalence in Ethiopia which could benefit your discussion (see Bisetegn et al., 2021). Heilegebriel et al., 2021 gives more recent data for Northwestern Ethiopia than Worku et al., 2014 cited by the author in line 239 and the 2013 study cited in line 225.

Structure conforms to journal standards, discipline norm
Figures are relevant, moderate quality therefore requiring improvement as follows:
Figure 1: Arrows cutting through the words ‘Proportional allocation of study subjects by school’ makes the words less clear. This should be revised.
Figure 2: Placement of the x-axis label ‘positive’ and ‘negative’ coincides with the red 10-14 bar rather than all the bars in each category. Positive is also misspelt
Table 1 legend should define what A, B, C represents
Raw data has been supplied as per the journal policy

Experimental design

Original primary research within Scope of the journal.
Research question well defined, relevant & meaningful. It is stated how the research fills an identified knowledge gap.
Rigorous investigation performed to a high technical & ethical standard. Clarification is needed about enrolment of minors: Were they given adequate information for them to give verbal assent?
Almost 70% of the enrolled participants were male. The authors should at least explain this was the case and how it affects the findings.
Methods described with sufficient detail & information to replicate. The authors are however not clear if infection intensity S. mansoni was determined from the Kato-Katz results, and how this was calculated.
Additionally, how many slides were prepared from each stool sample? This needs to be stated to improve the methods section

Validity of the findings

Impact and novelty not assessed.
Authors have provided underlying data which are well analysed.
The authors mention swimming was reported among 39.2% of the participants, but don’t state where the swimming was taking place for it to be relevant (line 190).
Climate change is mentioned as a contributing factor to increased prevalence of S. mansoni, yet no evidence is provided to support this. Is the prevalence reported in this study an increase? From what levels? (line 231-236).
Conclusions are well stated, linked to original research question & limited to supporting results.

Reviewer 3 ·

Basic reporting

The authors provide a useful resource for understanding factors that influence the prevalence of schistosomiasis in Kersa. The introduction lays out the schisto problem well from a socio-economic point of view. I think what can be improved is a more explicit rendering of how this study provides a tangible resource for addressing schisto (essentially further developing lines 87-92).

The MS would be greatly improved with grammar revision. I have highlighted some examples in the general comments, but I am sure I missed some.

Each of the factors in the “characteristics” column of table 1 do not exist in a vacuum. Do the authors have sense to which combinations of factors (e.g., shoe wearing and household latrine use) is predictive of infection? It complicates the analyses immensely, but I think it would be important to at least comment on multiple and simultaneous factors that could together be a stronger predictor of infection.

A suggestion: A strength of this study is that its results are directly comparable to other studies that have used a similar design (e.g., citations 22-25 and others). While the discussion lays out some general comparisons, it is difficult to understand fully the trends that are (and are not) comparable between studies. I think adding a table of the commonalities/differences from your study and others as well would provide a more holistic understanding of infection factors. Each column would be a different paper (reference) and each row would be a simple binary for each factor of yes/no for statistical significance for the associated factor. Essentially organizing the discussion into an easily digestible table where trends are easy to spot. This idea might be its own small paper and outside of the scope of your own goals, but I think adding the table would provide a much broader understanding of the schisto problem. Just a suggestion!

Experimental design

The authors do a good job of designing the experimental protocol including determining an appropriate sample size for the study. Their basis for participant exclusion is valid and efforts to randomize sampling appropriate.

The authors did an admirable job in reporting their efforts to accurately replicate the stool collection protocol for each individual student by training data collectors and interviewers.

On line 240 an expanded national deworming program is mentioned but not expanded upon. The authors suspect that the lower prevalence of S. mansoni infection, compared with similar studies, might be a contributing factor. For this study, I am curious about the extent to which participants that had been treated for schistosomiasis (deworming via PZQ) would be something to consider for total prevalence. I see that in the questionnaire (question 21) asks whether the individual had been treated in the past three months. Can the authors comment/speculate on the extent to which PZQ deployment is driving infection prevalence as a whole?

Validity of the findings

The collection method, careful participant exclusion, and determination of statistical significance is appropriate to have confidence in the findings. I can’t comment directly on the validity of the stats, but it seems to me that the strongest predictor of infection was bathing in rivers and ponds (only 2 individuals that did not bathe in rivers were positive for mansoni). This result isn’t really given much attention in the text.

Additional comments

Line 105: What is the reason for only focusing on children and not adults as well?
Line 177-182: Not clear what the relevance of reporting parental occupations is.
Table 1. Unclear what is the significance of sub-columns A, B, and C. Adding an explanation in the legend would clear this up.
Italicized genus species names not consistent throughout (e.g., line 116, 117, 194, 245)

---

## Round 0.2 · Minor Revisions

The authors addressed the main concerns of the reviews. However, the revised manuscript still deserves attention. Please provide point-to-point responses according to the comments made by the Reviewers in the new version of your manuscript. I agree with Reviewer #1 that the discussion could be enhanced with a more detailed analysis of the main findings reported in the study in comparison to the data already published in the literature. The English language and writing still need to be improved.

**Language Note:** The Academic Editor has identified that the English language must be improved. PeerJ can provide language editing services - please contact us at [email protected] for pricing (be sure to provide your manuscript number and title). Alternatively, you should make your own arrangements to improve the language quality and provide details in your response letter. – PeerJ Staff

·

Basic reporting

While the overall flow and grammar of the manuscript has been improved substantially, there are still areas needing to be revised to make the message clear. For example, the opening statement of the abstract implies that schistosomiasis is a disease and a parasite at the same time (line 21-22).
Other unclear statements include lines 25-26. The phrase 'protected water' (line 46) is unclear. Some of the edits affect the grammar therefore need reviewing, such as line 52-53, 66, 96-98, 124-126, (repetition of school children), 146-149 (repetition), spelling for Kao-Katz, using both upper case and lower case K is common. Lines 211-212, gave mean range and age, not clear what the author means.
Line 238, prevalence of 50 (62.5%), what does the 50 represent?.

In General, there is still quite a bit of work needed to improve the English language in the entire manuscript.

Experimental design

No comment

Validity of the findings

No mitigation measures are suggestion for the weaknesses cited by authors, including the use of a a single stool sample. Why were egg intensitis not determined during the reading of the slides? This is a major deficiency for Kato-Katz. It is also not accurate to state that Kato-Katz is the best diagnostic technique for use in areas of low prevalence as stated in line 167-168.

Additional comments

The discussion can do with some improvement, as the authors mostly cite several other studies for every finding that support/do not support their findings without any effort to analyse each study cited and it's comparability with their study. It would also be helpful to discuss findings and their implications locally and beyond.

Reviewer 3 ·

Basic reporting

The authors have adequately addressed my primary concerns with the manuscript. Additionally, it appears that the valid criticisms made by other reviewers have been addressed well. Specifically, much clarity has been added to describing the experimental design (e.g., Table 1 being replaced by Figure 1) and some of the reasoning behind chosen metrics make sense after the added detail. The grammar has been improved making the MS more readable. There are a few edits that should be addressed that are very minor:

There are still issues with italics for genus/species names (e.g., lines 82, 219, 325, 340).

Some of the spacing between words need to be looked at (e.g., line 222: "10 to14 years") and double spaces after periods throughout.

Given the minor edits needed I don't feel the need to do another revision. As I said in my previous review, this work is a useful study describing schisto prevalence in Ethiopia and I look forward to seeing it published.

Experimental design

The experimental design was not significantly altered from the previously reviewed manuscript.

Validity of the findings

The validity of findings were not significantly altered from the previously reviewed manuscript.

---

## Round 0.3 · accepted · Accept

After reviewing the revised manuscript, I confirm that all of the reviewers' comments have been effectively addressed.